# Social determinants of self-medication with leftover antibiotics in Lebanese households: A cross-sectional study

Reve Khaddaj[1]*, Pascale Salameh[1,2,3], Amal Al-Hajje[1,2], Julia Bou Dib[4], Joumana Yeretezian[4], Michele Cherfane[5], Reham Kotb[6], Diana Nakhoul[1], Reem Awad[1], Katia Iskandar[2,4,7,8]

**1** Faculty of Pharmacy, Lebanese University, Beirut, Lebanon, **2** INSPECT-LB (Institut National de Santé Publique, d'Épidémiologie Clinique et de Toxicologie-Liban), Beirut, Lebanon, **3** Department of Primary Care and Population Health, University of Nicosia Medical School, Nicosia, Cyprus, **4** Higher Institute of Public Health (ISSP), Saint Joseph University of Beirut, Lebanon, **5** Gilbert and Rose Marie Chagoury School of Medicine, Lebanese American University, Byblos, Lebanon, **6** Environmental and Public Health Department, College of Health Sciences, Abu Dhabi University, Abu Dhabi, UAE, **7** Department of Pharmaceutical Sciences, School of Pharmacy, Lebanese International University, Beirut, Lebanon, **8** Faculty of Public Health-Section 2 (CERIPH), Lebanese University, Fanar, Lebanon

* reve-khaddaj@hotmail.com

## Abstract

### Background

Self-medication with leftover antibiotics (LA) is a global health crisis, particularly in contexts of economic and political instability. This study examines the social determinants of health (SDOH) influencing LA use among Lebanese households, focusing on how individuals are born, grow, live, and work.

### Methods

A cross-sectional study was conducted among 368 Lebanese adults to collect data on socio-demographic characteristics, self-medication practices related to LA, and key SDOH such as socioeconomic status (SES), perceived discrimination in medical settings (DMS), political and economic instability, drug shortages, and trusted sources of health information. Data were analyzed using bivariate tests and multivariable logistic regression.

### Results

Bivariate analysis showed that LA use was significantly associated with male gender (45.0% vs. 33.3%, $p = 0.024$), lower educational levels (58.7% vs. 35.1%, $p = 0.002$), presence of chronic disease (54.7% vs. 32.2%, $p < 0.001$), older age ($p = 0.028$), and a higher household crowding index ($p = 0.018$). LA use was also more prevalent among participants impacted by political instability, economic crisis, and drug

**Data availability statement:** All relevant data are publicly available from the figshare repository (https://doi.org/10.6084/m9.figshare.28465301).

**Funding:** The author(s) received no specific funding for this work.

**Competing interests:** The authors have declared that no competing interests exist.

**Abbreviations:** AMR: Antimicrobial Resistance; CI: Confidence Interval; DMS: Discrimination in Medical Settings; LA: Leftover Antibiotic; LMIC: Lower- and Middle-Income Countries; SD: Standard Deviation; SES: Socioeconomic Status; SDOH: Social Determinant of Health; SM: Self-Medication; WHO: World Health Organization.

shortages (all $p<0.001$). In multivariable logistic regression, chronic disease (OR = 2.711, $p=0.002$), economic crisis (OR = 2.013, $p<0.001$), and prior experience with the same illness (OR = 4.085, $p<0.001$) were identified as significant predictors for LA use.

## Conclusion

These findings show the critical role of socio-economic instability, healthcare access challenges, and experiential factors as key SDOH driving LA practices. The study highlights the urgent need for multi-sectoral interventions addressing economic hardship, improving healthcare access, and enhancing public awareness to mitigate LA misuse and its contribution to antimicrobial resistance in Lebanon.

## 1. Introduction

Antibiotics have revolutionized medical discoveries of the 20th century and shifted the treatment paradigm of bacterial infections [1–4]. While these drugs continue to prevent an estimated 700,000 deaths annually from bacterial infections, [5] their effectiveness faces an unprecedented challenge through antimicrobial resistance (AMR). With the rapid global spread of AMR, the World Health Organization (WHO) has classified this ongoing crisis as one of the top three public health threats and leading causes of death worldwide [6]. Recent studies indicated that AMR-related deaths reached approximately 1.27 million in 2019, while nearly 5 million deaths were associated with drug-resistant infections [6]. For decades, the exploitation of these drugs has remained widespread across both developed and developing countries, where they are freely used in human medicine and animal husbandry, serving therapeutic, metaphylactic, and prophylactic purposes in livestock [2,3,7].

A significant driver of AMR is the inappropriate use of antibiotics, predominantly through self-medication (SM) with leftover antibiotics (LA) [8,9]. Defined as unused portions of prescribed antibiotics that patients retain after completing their treatment regimen, LA emerges from over-prescription and incomplete adherence to prescribed courses [10]. The global landscape of antibiotic use shows alarming trends, with consumption surging by over 20% since 2016, particularly in lower- and middle-income countries (LMICs) [11]. More concerning is that 80% of antibiotics are used outside official healthcare settings, with 20–50% of these uses being deemed inappropriate [11]. The practice of self-medication with LA varies significantly across regions, ranging from 77.5% in Saudi Arabia [12] to 6% in the United Kingdom [13], with intermediate rates of 26.69% in Bangladesh [8] and 44.1% in South America [14].

Beyond its significant role in driving AMR, LA use is a detrimental practice to individual health and broader public health systems [10,15]. This practice is often associated with multiple health risks, including the use of incorrect or expired medications, increased risk of adverse drug reactions, inappropriate drug selection, incomplete courses of treatment, delayed recovery, and higher risk of complications [16,17].

To better understand the underlying drivers of this public health threat, exploring the underlying associated social determinants of health (SDOH) is a pressing need. [18]. The WHO defines SDOH as "the non-medical factors that influence health outcomes, including the environments in which individuals live, work, and age, as well as the broader social, economic, and political systems that shape these conditions" [19,20]. These determinants include various factors such as education, income level, healthcare accessibility, cultural beliefs, and health literacy. [18–20]. These determinants contribute to health inequities, where health outcomes vary unfairly across socioeconomic groups. Research suggests that SDOH can account for up to 55% of health outcomes, highlighting their importance on healthcare services or personal lifestyle choices. [19,21–23].

The COVID-19 pandemic, compounded by Lebanon's persistent political and economic crises, severely strained the healthcare system, exposing critical gaps in disease control, medication access, and public health infrastructure [24,25]. Although the pandemic initially heightened public awareness regarding infection prevention and antimicrobial resistance, the post-pandemic period witnessed a regression in appropriate antibiotic practices, as systemic challenges and socioeconomic hardships were endured, perpetuating misuse behaviors such as LA use. Lebanon exemplifies the intersection of economic hardship and health system fragility within this global context, serving as a compelling case study of antibiotic misuse [25,26]. The nation's healthcare system, already burdened by conflict and financial collapse, faces severe challenges, including currency devaluation, medication shortages, and healthcare professional migration. [21,22,27]. Recent studies in Lebanon indicate that 23.89% of individuals keep leftover antibiotics at home, with 13.11% expired [10]. The storage, sometimes stockpiling, and subsequent use of LA represents a complex public health challenge stemming from multiple interconnected factors, including healthcare accessibility, prescribing practices, and patient behavior [10,15].

While global research has explored the relationship between SDOH and self-medication with LA [10,28], such as education and knowledge levels of antibiotic use and AMR [29–31], socioeconomic situation [32,23,33,34], access to healthcare services [10,32,35,36], there is a critical gap remaining in understanding how these factors specifically influence leftover antibiotic use in the Lebanese context. As a middle-income nation grappling with complex healthcare challenges, Lebanon's case highlights the urgent need for targeted research.

This study explores how SDOH shapes self-medication practices with LA in Lebanon, offering critical insights to inform global efforts to combat AMR and support policy development for sustainable healthcare interventions in the region and beyond.

This gap is particularly significant given Lebanon's unique position as a middle-income country experiencing severe economic challenges and a complex health system infrastructure [22].

The primary objective of our study is to identify the social determinants that contribute to the use of leftover antibiotics among the Lebanese population.

## 2. Methods

### Study design

A cross-sectional prospective study was conducted from August 22, 2024, to October 02, 2024, to comprehensively evaluate the SDOH of LA use in Lebanon.

### Participants

Eligible participants were Lebanese adults, 18 years of age and above, residing in Lebanon and have internet access.

### Choice of SDOH

The SDOH selection was guided by the WHO's conceptual framework [19], which provides a structured approach to understanding how various social factors influence health outcomes through both structural and intermediary

determinants. The selected SDOH included traditional socioeconomic indicators (education, household crowding, economic status) and context-specific elements reflective of Lebanon's ongoing crisis (political instability, drug shortages, healthcare discrimination). The inclusion of discrimination and healthcare access measures was especially pertinent given Lebanon's complex sectarian landscape [37] and their documented impact on healthcare delivery and outcomes [38–40]. Incorporating family influence and information sources intended to capture the role of social networks in Lebanese healthcare decision-making, an aspect often underexplored in traditional SDOH frameworks, but particularly relevant in Middle Eastern contexts. This integrated approach enabled a comprehensive examination of how multiple layers of social determinants influence medication behaviors in a fragmented healthcare system [22].

## Questionnaire

The questionnaire consisted of three parts, in addition to the introduction to the survey that explained the study topic, objectives, and a statement of consent. The participants were asked if they had ever self-medicated with antibiotic leftovers (Appendix 1).

The first part included the socio-demographic characteristics of participants, such as age, gender, marital status (single and ever married which comprises married, divorced, and widowed), education level (having a university degree or not), health education or not, having children, region of living, distance from their home to the nearest pharmacy (near is considered for a distance lower than 5 Km and far is considered for a distance of 5 Km or more), having or not chronic diseases, area of residence (urban or rural), and the household crowding index, a measure used to assess residential overcrowding, calculated by dividing the number of people living in the house by the total rooms excluding kitchens and bathrooms [41].

The second part focused on examining the practice of self-medication with LA and consisted of two sections:

A. Sources of LA: The participants who self-medicated with LA (answered "yes") were provided four options to indicate the sources of antibiotic leftovers used, including friends, neighbors, family, and household leftovers.

B. Reasons for self-medication or avoidance: The participants who self-medicated with LA (answered "yes") were given 16 predefined answers to identify their motivations to self-medicate. All participants (who answered "yes" or "no" on the question related to LA use), were provided 6 predefined options to capture the reasons that prevent them from self-medication. These questions were inspired by similar studies [42,43].

The third part encompassed the SDOH selected based on their direct relevance to the research question and the socio-political context of Lebanon, with consideration of practical constraints. This part consisted of 5 sections:

1- The Socioeconomic Status Scale (SES scale) [44], designed to capture respondents' financial strains, consists of 8 items. Participants were asked to rate their financial stress over the past month on a scale from 1 to 10, covering areas such as essential expenses and financial stability. Higher scores indicated lower financial stress and better financial well-being. The total possible score ranged from 8 to 80. The scale demonstrated internal consistency (Cronbach's alpha = 0.935).

2- The Discrimination in Medical Setting Scale (DMS scale) [45] was used to evaluate patients' self-reported experiences of discrimination in healthcare environments. The scale includes 7 items rated on a 5-point Likert scale (1 = Never to 5 = Always), with higher scores reflecting greater perceived discrimination. The total possible score range is 7 to 35. The scale demonstrated strong internal consistency (Cronbach's alpha = 0.89). Participants reported the frequency of discrimination experiences related to their clinical conditions during medical interactions.

3- The socio-economic and political conditions in Lebanon, including factors such as ongoing political conflicts at the time of data collection, political instability, drug shortages, and the ongoing economic crisis. The assessment aimed

to capture the contextual challenges influencing healthcare behaviors during periods of national crisis. Each item was rated on a 5-point scale (1 = No influence, 5 = very strong influence), where higher scores indicate a greater perceived impact of socioeconomic and political factors on their SM behavior.

4- Trusted source of information consisted of 9 options adapted from previous studies [46–49]. Respondents were asked to indicate whether they trusted each source, using a yes/no format. It was formulated to assess the source of information that participants relied on when using LA without a doctor's prescription.

5- Level of knowledge of the participants about antibiotic use, antibiotic resistance, and LA use risks: it was divided into three sub-sections, each using a standardized 5-point Likert scale ("strongly disagree" to "strongly agree"). For all sub-sections, the scoring methodology followed these principles. Respondents rated their agreement on a 5-point Likert scale ranging from "strongly disagree" to "strongly agree." Correct responses were assigned one point based on the statement type. For true statements, selecting "agree" or "strongly agree" was correct, while for false statements, selecting "disagree" or "strongly disagree" earned a point. Neutral responses were considered incorrect. Reverse coding was applied where necessary for negatively phrased statements. Participants scoring 70% or higher were classified as having good knowledge [50]. The sub-sections were structured as follows:

- Knowledge of antibiotic use sub-section included 18 statements adapted from previous studies [30,51–55]. It focuses on antibiotic effects, use, and side effects.

- Knowledge of antibiotic resistance sub-section comprised 9 statements adapted from previous research [30,51–53]. It focuses on AMR spread mechanisms, global impact, personal health implications, transmission patterns between humans and animals, and contributing factors to resistance development.

- Knowledge of risks associated with LA use sub-section included 12 statements adapted from previous studies [43,56]. It assesses the knowledge about risks and potential dangers associated with SM using leftover antibiotics, including understanding inappropriate drug use, safety concerns, diagnostic errors, adverse effects, therapeutic failures, and economic consequences.

## Data collection

Data were collected using an online questionnaire formulated in English and created on Google Forms, a cloud-based survey powered by Google™. A forward translation was first performed from English into Arabic and then back-translated into English. The two English versions were compared, with minor discrepancies corrected by consensus between the translators and the authors. A pilot test of the questionnaire included 15 participants from different age groups, genders, socio-economic and educational backgrounds to identify any potential ambiguities in language or expressions. Subsequent feedback prompted necessary revisions for clarity and cultural relevance. The link to the questionnaire was shared using various social media platforms, including WhatsApp, Facebook, Instagram, and LinkedIn. Data were collected using the snowball sampling technique [57].

## Ethical approval

This study, approved by the Institutional Review Board at the Lebanese International University under code 2024ERC-020-LIUSOP, adhered to the Declaration of Helsinki [58]. Before filling out the online survey, participants were briefed about the study objectives and their right to withdraw at any time. Written informed consent was obtained from all participants by including a consent statement at the beginning of the online survey. Participants were required to read the statement and click a checkbox indicating their agreement to participate, which enabled them to proceed. Participants did not receive any financial

reward for their participation. The online survey was anonymous and voluntary. Collected data were encrypted, stored in password-protected computers, and presented as de-identified electronic files in Microsoft Excel and SPSS.

### Inclusivity in global research

Additional information regarding the ethical, cultural, and scientific considerations specific to inclusivity in global research is included in the Supporting Information S1 File.

### Sample size calculation

The minimal sample size was calculated using Epi-Info version 7.2.6. Based on a previously published study in Lebanon [10]. The prevalence of LA use among the Lebanese population was 23.89%. The minimum necessary sample was n = 279 participants, considering an alpha error of 5% and a power of 95%. A minimum sample of 300 participants was targeted to account for potential missing values.

### Statistical analysis

Data were analyzed using SPSS version 26. Descriptive statistics were performed to summarize the sociodemographic characteristics, the impact of the Lebanese crisis on the decision to use leftover antibiotics, the SDOH, and knowledge regarding antibiotics, antibiotic resistance, and risk of LA use. Continuous variables were summarized using means, standard deviations (SD), and medians, while categorical variables were presented as percentages.

Bivariate analysis was conducted to examine the association between leftover antibiotic use (dependent variable) and other independent variables. For categorical variables, Chi-square tests were used, and for continuous variables, the Student's t-test was applied. A p-value < 0.05 was considered statistically significant.

Logistic regression was employed with the use of leftover antibiotics as the dependent variable. Independent variables with a p-value < 0.2 from the bivariate analysis were included in the model using the Enter method. The Omnibus test was performed to assess model significance, while the correlation matrix was examined to check for collinearity. The Hosmer and Lemeshow tests were used to evaluate the model's goodness-of-fit, and Nagelkerke $R^2$ was reported to determine the explanatory power of the model.

## 3. Results

### Socio-demographic characteristics

Table 1 summarizes the socio-demographic characteristics of participants (N = 368) and their association with self-medication using LA. A significant association was observed with gender (p = 0.024), as males were more likely to self-medicate (45.0%) than females (33.3%). Education level also showed significance (p = 0.002), with participants holding a university degree reporting the lowest self-medication rate (35.1%). The presence of chronic disease was strongly associated with self-medication behavior (p < 0.001). Although marital status and parenthood showed no significant associations, ever-married participants (40.6%) and those with children (40.7%) exhibited higher self-medication rates. No significant relationship was found with the field of study, region category, area of living, or proximity to a pharmacy. However, participants living near a pharmacy reported higher self-medication (39.2%) than those farther away (24.1%, p = 0.108). Age (45.50 ± 13.599 years in non-LA users, and 48.72 ± 13.618 years in LA users, p = 0.028) and household crowding index (HCI: 0.904 ± 0.471 in non-LA users, and 1.061 ± 0.666 in LA users, p = 0.018) were significantly associated with LA use.

### Practice of self-medication

Fig 1 shows that the primary source of LA used is household antibiotics (76.3%) followed by those borrowed from family (32.4%). Relying on neighbors and friends as a source of LA was the least (7.9% and 13.7% respectively).

**Table 1. Socio-demographic characteristics of participants (N = 368).**

**Self-medication with Leftover antibiotics (N = 368)**

| Variables | No n = 228 | Yes n = 140 | p-value |
|---|---|---|---|
| **Gender** | | | |
| Male | 82 (55.0) | 67 (45.0) | 0.024* |
| Female | 146 (66.7) | 73 (33.3) | |
| **Marital Status** | | | |
| Single | 63 (70.0) | 27 (30.0) | 0.071 |
| Ever married | 165 (59.4) | 113 (40.6) | |
| **Have Children** | | | |
| No | 81 (67.5) | 39 (32.5) | 0.128 |
| Yes | 147 (59.3) | 101 (40.7) | |
| **University Degree** | | | |
| No | 19 (41.3%) | 27 (58.7%) | 0.002* |
| Yes | 209 (64.9%) | 113 (35.1%) | |
| **Field of Study** | | | |
| Health | 46 (66.7) | 23 (33.3) | 0.371 |
| Non-health | 182 (60.9) | 117 (39.1) | |
| **Suffering From Chronic Disease** | | | |
| No | 185 (67.8) | 88 (32.2) | <0.001** |
| Yes | 43 (45.3) | 52 (54.7) | |
| **Region Category** | | | |
| Beirut | 38 (61.3) | 24 (38.7) | |
| North | 39 (69.6) | 17 (30.4) | 0.474 |
| Mount Lebanon | 103 (61.7) | 64 (38.3) | |
| Beqaa | 27 (64.3) | 15 (35.7) | |
| South | 21 (51.2) | 20 (48.8) | |
| **Area of Living** | | | |
| Urban | 151 (62.1) | 92 (37.9) | 0.920 |
| Rural | 77 (61.6) | 48 (38.4) | |
| **Distance From Home to Nearest Pharmacy** | | | |
| Near | 206 (60.8) | 133 (39.2) | 0.108 |
| Far | 22 (75.9) | 7 (24.1) | |
| | **Mean (SD)** | | |
| **Age (years)** | 45.500 ± 13.599 | 48.720 ± 13.618 | 0.028* |
| **House Crowding Index (HCI)** | 0.907 ± 0.471 | 1.061 ± 0.666 | 0.018* |

\* Indicates significant value (p < 0.05); \*\* Indicates highly significant value at a level of 1% (p ≤ 0.001);

Near: < 5 km; Far ≥ 5 km; SD: Standard Deviation; Ever Married: Married, widowed, or divorced.

Fig 2 indicates that participants cited various reasons for engaging in this practice. The most frequent reasons include having a known disease (87.9%) and experiencing minor ailments (89.2%), followed by the availability of leftover antibiotics (70.7%) and convenience (69.3%). Knowledge about medications (85.7%) and inadequate time to visit a doctor (73.6%) were also predominant factors. Additionally, quick relief (83.6%) and saving time (58.6%) were notable motivations, while privacy (26.4%) and distrust in doctors (12.9%) were reported by fewer participants.

Fig 3 compares the reasons that prevent participants from self-medicating with LA between those who self-medicate with LA and those who do not. Among non-users of LA, the most common reasons include the belief that there is no

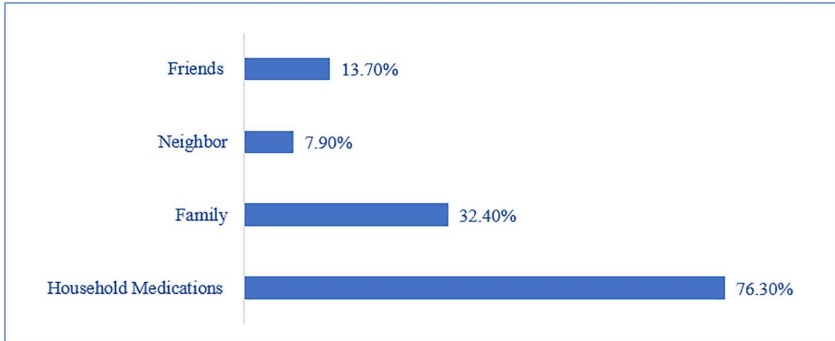

**Fig 1. Source of leftover antibiotic used.**

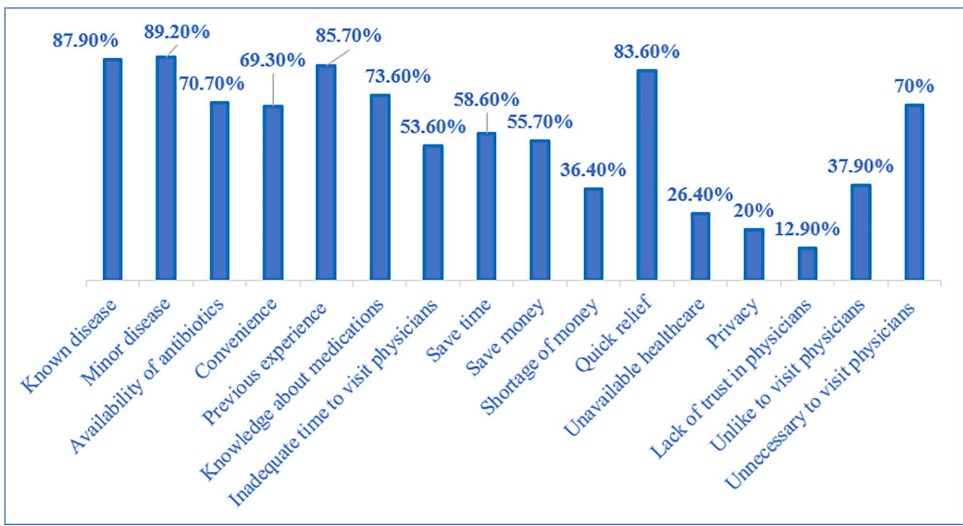

**Fig 2. Reasons encouraging self-medication with antibiotic leftovers.**

need to self-medicate (70.6%), concerns about the risk of side effects (64.9%), and a preference for medical consultation (63.8%). Additionally, 63.4% of non-users trust doctors for a better diagnosis, while 58% cited a lack of knowledge as a reason to avoid SM. In contrast, these concerns were less common among those who self-medicate, with only 29.4% seeing no need for self-medication, 35.1% worried about side effects, and 36.2% preferring medical consultation. Similarly, fewer self-medication users trusted a doctor's diagnosis (36.6%) or were aware of their lack of knowledge (42.3%). Other safety concerns were also more frequently reported by non-LA users (60%) compared to LA users (40%).

## Social determinant of health

Table 2 indicates a significant difference observed in DMS scale scores, with those who used LAs scoring lower than non-users (p = 0.029). Participants' perceptions of the political situation, drug shortage, and economic crisis in Lebanon were significantly associated with LA use (p < 0.001). Additionally, the ongoing political conflict in Lebanon was significantly associated with LA use (p = 0.047). Moreover, those who used LAs had significantly lower knowledge of antibiotics compared with non-users (p = 0.005). No significant associations were found between LA use and the SES scale (p = 0.130),

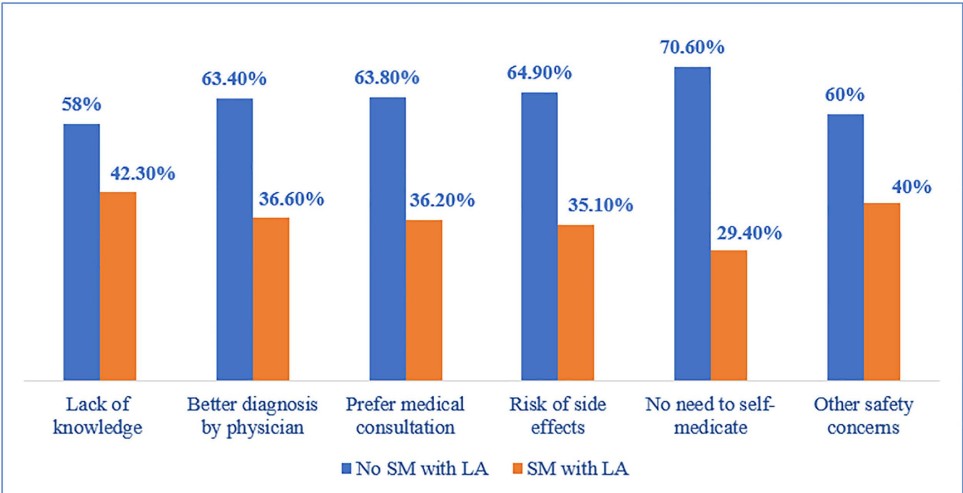

**Fig 3. Reasons for avoiding leftover antibiotic use between participants self-medicating and not self-medicating with leftover antibiotics.**

**Table 2. Social determinants of leftover antibiotic use.**

| Leftover Antibiotic Use | | | |
|---|---|---|---|
| **Variables** | **No n = 228** | **Yes n = 140** | **p-value** |
| **Mean (SD)** | | | |
| **SES Scale** | 38.065 ± 17.809 | 35.250 ± 16.366 | 0.130 |
| **DMS Scale** | 31.017 ± 4.225 | 29.892 ± 5.061 | 0.029* |
| **Lebanese socio-economic and political context** | | | |
| Political Situation | 1.910 ± 1.251 | 2.540 ± 1.547 | <0.001** |
| Drug Shortage | 2.270 ± 1.428 | 3.250 ± 1.541 | <0.001** |
| Economic Crisis | 2.010 ± 1.282 | 3.050 ± 1.485 | <0.001** |
| Ongoing Political Conflict | 1.790 ± 1.265 | 2.080 ± 1.368 | 0.047* |
| **Level of knowledge** | | | |
| AMR | 4.350 ± 2.831 | 4.450 ± 2.844 | 0.734 |
| Antibiotics | 11.050 ± 4.618 | 9.700 ± 4.039 | 0.005* |
| The risks of LA use | 9.12 ± 2.93 | 8.68 ± 3.05 | 0.200 |

*Indicates significant value (p < 0.05); **Indicates highly significant value at a level of 1% (p ≤ 0.001).

SD: Standard Deviation; SES: Socioeconomic Status; DMS: Discrimination in Medical Setting; AMR: Anti-microbial Resistance; LA: Leftover Antibiotics.

knowledge about AMR (p = 0.734), and knowledge about the risks of LA use (p = 0.200). Participants scoring 70% or higher were classified as having good knowledge [50] which means that a score considered good knowledge is 6 for AMR knowledge, 12 for antibiotics knowledge, and 8 for risk of LA use knowledge.

Table 3 showed a significant association between LA use and participants' previous experience of the disease (p < 0.001), with individuals who had prior disease experience being more likely to use leftover antibiotics (49.8%) than those without such experience (19.6%). Additionally, reliance on a family member's opinion was significantly associated with LA use (p < 0.001), as 51.7% of those influenced by family opinions reported LA use compared to 31.6% who did not. Other factors, such as the role of pharmacists, academic knowledge, internet usage, reading materials, social media, mass media, and artificial intelligence, showed no statistically significant association with LA use (p > 0.05).

**Table 3. Trusted source of information about leftover antibiotic use.**

| Leftover antibiotic use | | | |
|---|---|---|---|
| | No n = 228 | Yes n = 140 | |
| Variables | Frequency (%) | | p-value |
| **Pharmacist** | | | |
| No | 12 (50.0) | 12 (50.0) | 0.212 |
| Yes | 216 (62.8) | 128 (37.2) | |
| **Previous Experience with The Disease** | | | |
| No | 115 (80.4) | 28 (19.6) | <0.001** |
| Yes | 113 (50.2) | 112 (49.8) | |
| **Academic Knowledge** | | | |
| No | 99 (59.3) | 68 (40.7) | 0.335 |
| Yes | 129 (64.2) | 72 (35.8) | |
| **A Family Member's Opinion** | | | |
| No | 171 (68.4) | 79 (31.6) | <0.001** |
| Yes | 57 (48.3) | 61 (51.7) | |
| **Internet** | | | |
| No | 117 (66.9) | 58 (33.1) | 0.065 |
| Yes | 111 (57.5) | 82 (42.5) | |
| **Reading Materials** | | | |
| No | 168 (64.9) | 91 (35.1) | 0.076 |
| Yes | 60 (55.0) | 49 (45.0) | |
| **Social Media** | | | |
| No | 198 (62.9) | 117 (37.1) | 0.386 |
| Yes | 30 (56.6) | 23 (43.4) | |
| **Mass Media** | | | |
| No | 204 (62.2) | 124 (37.8) | 0.787 |
| Yes | 24 (60.0) | 16 (40.0) | |
| **Artificial Intelligence** | | | |
| No | 216 (62.4) | 130 (37.6) | 0.460 |
| Yes | 12 (54.5) | 10 (45.5) | |

**Indicates highly significant value at a level of 1% (p ≤ 0.001).

## Multivariable analysis

Table 4 showed the results of the logistic regression, which revealed several significant predictors of the outcome (the use of LA). Relying on previous disease experience as a source of information emerged as the strongest predictor ($\beta = 1.407$, 95% CI [2.242;7.442], p<0.001), with an odds ratio indicating that individuals with prior experience with the disease were over four times more likely to experience the outcome. The presence of a chronic disease significantly increased the likelihood of the outcome ($\beta = 0.997$, 95% CI [1.443;5.094], p=0.002), while the economic crisis also demonstrated a substantial effect ($\beta = 0.700$, 95% CI [1.383;2.931], p<0.001). Drug shortage showed marginally significant associations ($\beta = 0.219$, 95% CI [0.980–1.582], p=0.073). Notably, demographic factors, including gender, marital status, having children, age, crowding index, and education level, did not significantly predict the outcome. Similarly, the discrimination scale, SES scale, knowledge about antibiotic use and risks of LA use, family opinion, pharmacist, Internet, and reading materials showed no significant associations with the outcome. Moreover, the political conflicts in Lebanon and the region, and the difficult political situation in Lebanon, did not show a significant correlation with LA use.

**Table 4. Logistic regression using self-medication with LA as the dependent variable.**

| Variable | B | Odds ratio | p-value | 95% CI |
|---|---|---|---|---|
| Gender (male*/female) | −0.261 | 0.770 | 0.352 | 0.445-1.334 |
| Marital Status (single*/ever married) | −0.185 | 0.831 | 0.728 | 0.293-2.356 |
| Having Children (No*/yes) | 0.997 | 1.448 | 0.435 | 0.571-3.673 |
| Age | −0.007 | 0.993 | 0.593 | 0.969-1.018 |
| Household Crowding Index | 0.350 | 1.419 | 0.158 | 0.872-2.309 |
| University Degree (No*/yes) | −0.537 | 0.585 | 0.208 | 0.254-1.347 |
| Chronic Diseases (No*/yes) | 0.997 | 2.711 | 0.002** | 1.443-5.094 |
| SES Scale | 0.005 | 1.005 | 0.564 | 0.989-1.021 |
| Discrimination Scale | −0.020 | 0.980 | 0.535 | 0.921-1.044 |
| Political Situation | −0.321 | 0.726 | 0.132 | 0.478-1.101 |
| Economic Crisis | 0.700 | 2.013 | <0.001*** | 1.383-2.931 |
| Drugs Shortage | 0.219 | 1.245 | 0.073† | 0.980-1.582 |
| Ongoing Political Conflict | −0.233 | 0.792 | 0.137 | 0.582-1.077 |
| Knowledge About Antibiotic Use | −0.036 | 0.965 | 0.252 | 0.908-1.026 |
| Knowledge About Risks Of LA Use | −0.040 | 0.960 | 0.354 | 0.881-1.046 |
| Pharmacist (No*/yes) | −0.553 | 0.575 | 0.285 | 0.209-1.584 |
| Previous Experience with The Disease (No*/yes) | 1.407 | 4.085 | <0.001*** | 2.242-7.442 |
| Family Opinion (No*/yes) | 0.155 | 1.167 | 0.595 | 0.660-2.065 |
| Internet (No*/yes) | 0.121 | 1.129 | 0.676 | 0.639-1.994 |
| Reading Material (No*/yes) | 0.168 | 1.183 | 0.596 | 0.635-2.206 |

*Reference variable;

**Indicates significant value (p < 0.05); ***Indicates highly significant value at a level of 1% (p ≤ 0.001); †Indicated marginally significant value (p < 0.10).
CI: Confidence Interval; LA: Leftover Antibiotics; SES: Socioeconomic Status

## 4. Discussion

This study examined the social determinants of LA use in Lebanon, revealing complex interactions among health-system, economic, and social factors. It showed a concerning 38% prevalence of LA use, significantly exceeding the 23.89% rate reported in a recent Lebanese study [10]. These findings position Lebanon between countries with lower prevalence, such as Turkey (34.2%) [59] and countries with higher prevalence, like Saudi Arabia (77.5%) [12]. The reliance on leftover antibiotics represents a coping mechanism, as documented in two Lebanese studies [10,60], where individuals navigate the dual pressures of financial constraints and inadequate healthcare access [61]. This adaptation to economic constraints presents a particular challenge for antimicrobial stewardship efforts, as individuals may prioritize immediate medication access over long-term resistance concerns [62]. This prevalence is particularly alarming given the spread of AMR in Lebanon [63] and its potential health and environmental impact in an AMR-endemic Middle-Eastern Mediterranean region [64].

Participants identified several reasons for self-medication in Lebanon, with findings showing that 73.6% of the participants self-medicated due to medication knowledge. This result is higher than results in other countries (48.8% in Iran [65], 42.3% in Serbia [66], and 23.59% in Nepal [49]). A total of 83.6% of participants indicated immediate relief-seeking behavior as a reason that encouraged them to self-medicate, a high rate exceeding those reported in other countries, with 28% in Saudi Arabia [12] and 19.1% in Nepal [49].Additionally, feedback from participants about reasons for avoiding LA use was also collected. Safety concerns were noted by 40% of LA users and 60% of non-users, while 70.6% of non-users cited no need for self-medication compared to 29.4% of LA users. These patterns reflect the complex interplay between insufficient regulatory oversight, compromised healthcare access, and economic constraints in shaping medication behaviors.

The predominant sources of LA reveal concerning patterns, with household antibiotics (76.3%) and family borrowing (32.4%) being the main sources. The high rate of household antibiotic storage may indicate several underlying issues. First, it could reflect inappropriate prescription practices, where patients are given larger quantities of antibiotics than needed, leading to leftovers. Second, it may point to poor adherence to treatment, with individuals stopping their medication early once symptoms improve or due to side effects. Third, some households might be intentionally saving antibiotics for future use, engaging in self-medication without medical guidance.

The pattern of borrowing medications within families reinforces the significant role of social networks in shaping access to treatment, often establishing informal pathways that bypass professional healthcare guidance. In particular, the reliance on family members for health-related decisions highlights the cultural context in which such behaviors occur. This is significant in our findings, where 51.7% of individuals who use LA reported relying on a family member's opinion ($p < 0.001$). Such results emphasize the deep trust placed in family advice, especially in societies characterized by strong familial bonds. While this trust fosters social support, it also increases the risk of inappropriate self-medication practices, reinforcing the urgent need for community-wide educational interventions aimed at promoting the rational use of antibiotics [67,68]. Furthermore, the widespread availability of LA raises concerns about over-prescription by healthcare providers. This may reflect inconsistent adherence to antibiotic stewardship guidelines, diagnostic uncertainty, or prescriber pressure to meet patient expectations. In Lebanon, such practices have been linked to limited enforcement of prescription regulations, inadequate continuing education, and the absence of standardized treatment protocols [69].

While this influence was not statistically significant in multivariate analysis, the qualitative patterns suggest that health decision-making in Lebanon occurs within a complex social framework that can either reinforce or challenge formal medical advice. This understanding is crucial for developing culturally appropriate interventions that acknowledge the role of family networks in medication practices, particularly in contexts where traditional healthcare systems may be compromised by economic and political instability. The reliance on family for health-related guidance as a trusted source of information may be particularly pronounced in cultures with strong familial ties, emphasizing the need for targeted educational interventions within these networks to mitigate inappropriate antibiotic use [67,68]. Conversely, other sources such as trust in pharmacists, academic knowledge, internet use, and social media showed no significant associations with LA use. This finding raises important questions about the effectiveness of these channels in shaping antibiotic consumption behaviors. For instance, despite the potential for pharmacists to provide valuable guidance on antibiotic use, their influence appears limited in this context. This could be attributed to barriers such as accessibility issues or a lack of proactive communication from healthcare professionals regarding proper antibiotic usage and disposal practices [70]. The absence of significant correlations with digital information sources also highlights a potential gap in public health messaging. As many individuals increasingly turn to online platforms for health information, it is crucial to assess the quality and reliability of such resources. The lack of engagement through these channels may suggest that public health campaigns need to adapt their strategies to better reach and educate populations about the risks associated with leftover antibiotics [43,71].

Personal health experiences emerged as the strongest predictors, particularly considering previous disease experience as a trusted source of information and chronic disease status. The results suggest that individuals with prior disease experiences develop increased self-efficacy in managing similar symptoms, potentially reducing their perceived need for professional healthcare advice [72]. Studies from Nepal [49] and India [68] reported similar patterns, suggesting that this may be a broader phenomenon in healthcare systems with limited regulatory oversight. This normalization of antibiotic self-management is particularly concerning in the context of AMR, as it may accelerate the development of resistant bacterial strains through inappropriate use patterns and create reservoirs of resistant organisms within communities [7].

Gender emerged as a significant factor, with males showing higher LA use (45%) compared to females (33.3%) ($p = 0.024$). This gender disparity warrants further investigation, potentially reflecting differences in healthcare-seeking behaviors, risk perception, or socioeconomic factors affecting healthcare access between men and women in the Lebanese context. These findings oppose previous studies in Lebanon that showed that females use LA more than males

because they are the care providers of their family, and because they appear to be proactive in their health-seeking behavior [10,40]. Similar studies conducted in Nepal [49] and Serbia [66] found no gender influence on self-medication practice. Additionally, the significant association of the household crowding index with LA use highlights the role of living conditions in medication behaviors. This finding suggests that household dynamics and living arrangements may influence medication sharing and storage practices, adding another layer to our understanding of social determinants.

Socio-economic and political stressors, particularly the ongoing economic crisis and widespread drug shortages, emerged as strong predictors influencing LA utilization. Interestingly, the SES scale, which typically includes indicators of individual financial strains, was not significantly associated with LA. While socioeconomic factors are often strong predictors of healthcare behaviors [73,74] as they reflect an individual's ability to access resources, our findings suggest a more complex dynamic in the Lebanese context. The severe devaluation of the Lebanese pound and its impact on purchasing power have made proper healthcare access increasingly challenging for many residents, pushing them toward potentially dangerous self-medication practices [33]. While numerous studies conducted in Lebanon [10], Saudi Arabia [46], and others [67,75] have documented the influence of the individual socio-economic status on health behavior, including antibiotic misuse, our study reveals a different pattern. The lack of significance in SES correlation may stem from the equalizing effects of Lebanon's ongoing economic crisis and drug shortages, which impact individuals across all socio-economic strata [33]. This crisis-specific context might overshadow traditional SES distinctions, as immediate situational factors such as drug shortage can take precedence over long-term socio-economic markers. Moreover, the SES scale may not fully capture critical variables relevant to LA use, such as trust in the healthcare system or informal medication sources, which may vary independently of SES. These findings suggest that LA use trends are shaped by a complex interplay of structural vulnerabilities and situational challenges, necessitating targeted interventions that address specific barriers like drug shortages and healthcare access limitations, rather than broader SES categorizations. The patterns reflect systemic challenges, including limited public health infrastructure, inadequate pharmaceutical supply chain regulation, and insufficient safety nets for economically disadvantaged populations.

The situation is further complicated by Lebanon's fragmented healthcare surveillance system and loose regulatory oversight of antibiotic dispensing, creating an environment conducive to inappropriate antibiotic use [22]. Addressing these issues requires a systematic approach that includes strengthening healthcare systems, improving the affordability and availability of medications, and implementing targeted public health interventions to reduce reliance on inappropriate self-medication practices.

Earlier national studies indicated that lower educational attainment was linked to higher rates of self-medication and antibiotic misuse [10,40]. Our findings align with previous research in this area, showing that participants with university degrees are less likely to self-medicate with LA (35.1% of university graduates use LA compared to 58.7% without university degrees, p = 0.008). Research from Saudi Arabia highlighted that inadequate knowledge about antibiotic resistance was prevalent among lower-educated individuals, leading to increased self-medication practices, mirroring our results where LA users had significantly lower knowledge about antibiotics [56]. Similarly, a cross-sectional study in Italy demonstrated that individuals with lower educational levels were more likely to engage in self-medication and exhibited poorer knowledge regarding antibiotic use, reinforcing the global trend observed in our research [17].

In our study, a statistically significant association was observed between education level and LA use, where 58.7% of LA users did not hold a university degree, compared to 35.1% who did (p = 0.002). This finding highlights the critical role of educational attainment in shaping medication practices. Individuals without higher education may have limited health literacy, making them less aware of the risks associated with antibiotic misuse, such as antimicrobial resistance and treatment failure. They are also more likely to rely on informal sources of health advice and may resort to using leftover medications due to economic constraints or lack of access to professional healthcare services [60]. This trend is consistent with the World Health Organization's report, which notes a strong correlation between low education levels and irrational antibiotic use in low- and middle-income countries. [32,64].

Our analysis revealed nuanced relationships between antibiotic knowledge and LA use. While knowledge of antibiotic resistance and LA use risks did not correlate significantly with LA use, there was a clear association between general knowledge about antibiotics and their usage patterns. Participants who reported using LA exhibited significantly lower knowledge scores compared with non-users. These findings suggest that enhanced knowledge about antibiotics correlates with more responsible usage behaviors, reducing the likelihood of retaining unused medications.

This pattern aligns with existing literature on education's role in antibiotic use. A Lebanese study found that inadequate understanding of antibiotic usage directly contributed to self-medication practices among various demographic groups [10]. Research from Cyprus reinforced this, showing how improved awareness and attitudes toward antibiotics could significantly reduce self-medication behaviors [52]. International studies have consistently shown similar patterns – from Serbia, where medical students' self-medication practices varied with their understanding of antibiotic protocols [66], to Saudi Arabia, where widespread misconceptions contributed to inappropriate use [30], and Ethiopia, where poor knowledge correlated with medication misuse [53]. A multicentric survey across 14 European countries further supported these findings, revealing that 84% of the public lacked knowledge regarding appropriate antibiotic use, with only 37% receiving antibiotic-related information in the previous year [76].

Another significant finding was the inverse relationship between perceived discrimination and LA use, contrasting with international literature. While other studies have frequently identified higher perceived discrimination as a barrier to healthcare access often resulting in increased self-medication practices [38,60,70], our findings in Lebanon's healthcare context suggest that individuals who perceive less discrimination have better access to formal healthcare services. This access could facilitate the accumulation of leftover antibiotics through formal prescriptions, potentially due to issues with prescription practices (i.e., dispensing without prescription, diagnostic uncertainty, lack of training and education), medication over-dispensing, or lack of patient adherence to prescribed regimens. The relationship between discrimination and LA use reveals a unique dynamic within Lebanon's sectarian healthcare system. LA users reported lower scores on the discrimination scale, suggesting that reduced discrimination correlates with greater access to medications. This aligns with the hypothesis that Lebanon's healthcare system, which often stratifies access based on socio-political affiliations and network ties, can unintentionally exacerbate inappropriate antibiotic usage by enabling those with better access to accumulate antibiotics [77]. These findings warrant further investigation into prescription and dispensing practices within Lebanon's healthcare system to better understand their implications for antibiotic stewardship. Addressing these structural issues, including over-prescription and inadequate monitoring of antibiotic use, is critical for minimizing antibiotic resistance risk and promoting more rational medication use.

## Limitations

Several methodological limitations warrant consideration when interpreting this study's findings. First, the cross-sectional design inherently precludes causal inference regarding the relationships between SDOH and LA use. Second, the snowball sampling technique used, while necessary given the challenging research context, may have introduced selection bias by potentially overrepresenting certain social networks and demographic groups. Although efforts were made to diversify recruitment across regions and socioeconomic strata, the age groups, socioeconomic groups, and areas of living were not equally represented. For this reason, the generalizability of our findings to the broader Lebanese population should be approached with caution. Third, the reliance on self-reported data presents another significant limitation, potentially introducing social desirability and recall biases. While the questionnaire was designed to minimize leading questions and ensure anonymity, participants' responses may have been influenced by their perceptions of socially acceptable behavior regarding antibiotic use. Fourth, the timing of data collection during a period of war, significant political conflicts, and economic crisis in Lebanon may have affected both response patterns and the representativeness of typical healthcare-seeking behaviors. Fifth, while our selection of social determinants was theoretically grounded in established SDOH frameworks, other unmeasured factors may influence LA use in the Lebanese context. Sixth, residual confounding

bias remains a potential limitation when examining the association between social determinants and LA use. Factors such as socioeconomic status, education, and healthcare accessibility may not always be fully captured or accurately measured, leading to an incomplete adjustment in statistical models. This can distort the observed relationship, either overestimating or underestimating the true effect of social determinants on LA use. For instance, if income level is not adequately accounted for, its residual influence may create a misleading association between education and LA use. Addressing this bias requires robust data collection and advanced analytical techniques to ensure a more accurate interpretation of findings. This limitation was addressed by doing a logistic regression.

This study was one of the first in Lebanon that examined the SDOH of LA use. It provided crucial insights regarding the impact of SDOH on the LA use, filling an important gap in the literature. Additionally, the large sample size studied was an important factor. Moreover, the completed data obtained (no missing data) is also an important strength of this study. Longitudinal studies are urgently needed to examine the temporal dynamics between social determinants and antibiotic use behaviors, particularly in the context of Lebanon's rapidly evolving socioeconomic landscape. Such studies would be invaluable in establishing causal relationships and understanding how changes in social conditions affect medication practices over time.

### Future directions

Future research should adopt longitudinal study designs to better establish causal relationships between SDOH and LA use. To address the biases associated with self-reported data, future studies should incorporate objective measures, such as pharmacy dispensing records or biomarker validation, to minimize recall and social desirability biases. Advanced analytical techniques, including structural equation modeling and mixed-methods approaches, are also warranted to unravel the complex and multifaceted interplay of SDOH influencing LA use behaviors. Furthermore, reinforcing surveillance systems for antibiotic consumption and resistance patterns is equally crucial, given Lebanon's position as a key player in the regional spread of AMR. Simultaneously, healthcare providers must be supported in managing patient expectations and prescribing behaviors, especially during economic hardships that can exacerbate inappropriate antibiotic use. Lastly, Lebanon's experience highlights the urgent need for a coordinated regional approach to combat AMR, recognizing that the SDOH influencing antibiotic misuse locally have broader implications for resistance patterns across neighboring countries.

## 5. Conclusion

This study highlights how LA use in Lebanon is driven by complex social determinants, including economic crises, healthcare system challenges, and prior disease experiences. Structural factors, like medication shortages, often outweigh individual factors such as education or knowledge about antibiotics, making LA use a coping mechanism rather than a simple personal choice. Addressing this issue requires a comprehensive approach that balances strengthening regulations and healthcare provider support with ensuring equitable access to medicines, especially in resource-limited and politically unstable settings.

## Supporting information

**S1 File. Inclusivity in global research.**
(DOCX)

**S2 File. Questionnaire – Appendix 1.**
(DOCX)

## Author contributions

**Conceptualization:** Reve Khaddaj.

**Data curation:** Reve Khaddaj.

**Formal analysis:** Reve Khaddaj, Katia Iskandar.

**Investigation:** Reve Khaddaj.

**Methodology:** Reve Khaddaj, Katia Iskandar.

**Project administration:** Reve Khaddaj, Katia Iskandar.

**Resources:** Reve Khaddaj.

**Software:** Reve Khaddaj, Julia Bou Dib, Joumana Yeretezian.

**Supervision:** Pascale Salameh, Amal Al-Hajje, Katia Iskandar.

**Validation:** Reve Khaddaj, Michele Cherfane, Reham Kotb, Diana Nakhoul, Reem Awad, Katia Iskandar.

**Visualization:** Reve Khaddaj, Julia Bou Dib.

**Writing – original draft:** Reve Khaddaj, Katia Iskandar.

**Writing – review & editing:** Reve Khaddaj, Pascale Salameh, Amal Al-Hajje, Michele Cherfane, Diana Nakhoul, Reem Awad, Katia Iskandar.

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
