## [Decision Letter · Decision Letter 0]

27 Jun 2025

PONE-D-25-10954Social Determinants of Self-Medication with Leftover Antibiotics in Lebanese Households: A Cross-Sectional Study.PLOS ONE

Dear Reve Rami Khaddaj,

Thank you for submitting your manuscript to PLOS ONE. After careful consideration, we feel that it has merit but does not fully meet PLOS ONE’s publication criteria as it currently stands. Therefore, we invite you to submit a revised version of the manuscript that addresses the points raised during the review process.

Kindly reply to all comments provided by both reviewers.Also note that it is important to have some discussions about the pandemic and post pandemic period in Lebanon and how it affected disease, public awareness and related issues to your work. Relevant work from Lebanon about the COVID19 pandemic include: Khoury 2020, El Deeb 2020 and Abou Hassan 2023.It is essential to provide detailed responses as well as suggested changes to be able to proceed with your manuscript.==============================

We look forward to receiving your revised manuscript.

Kind regards,

Omar El Deeb

Academic Editor

PLOS ONE

Journal Requirements:

3. Please include a complete copy of PLOS’ questionnaire on inclusivity in global research in your revised manuscript. Our policy for research in this area aims to improve transparency in the reporting of research performed outside of researchers’ own country or community. The policy applies to researchers who have travelled to a different country to conduct research, research with Indigenous populations or their lands, and research on cultural artefacts. The questionnaire can also be requested at the journal’s discretion for any other submissions, even if these conditions are not met.  Please find more information on the policy and a link to download a blank copy of the questionnaire here: https://journals.plos.org/plosone/s/best-practices-in-research-reporting. Please upload a completed version of your questionnaire as Supporting Information when you resubmit your manuscript.

4. Please ensure that you refer to Figure 1-3 in your text as, if accepted, production will need this reference to link the reader to the figure

Reviewers' comments:

Reviewer's Responses to Questions

**Comments to the Author**

1. Is the manuscript technically sound, and do the data support the conclusions?

Reviewer #1: Yes

Reviewer #2: Partly

2. Has the statistical analysis been performed appropriately and rigorously? 

Reviewer #1: Yes

Reviewer #2: No

3. Have the authors made all data underlying the findings in their manuscript fully available?

Reviewer #1: Yes

Reviewer #2: Yes

4. Is the manuscript presented in an intelligible fashion and written in standard English?

Reviewer #1: Yes

Reviewer #2: Yes

5. Review Comments to the Author

Reviewer #1: The research work is a relevant study which will potentially enhance scientific knowledge on the dynamics of non-prescribed antibiotic use, and provide evidence for antimicrobial stewardship especially in the LMICs where the study was conducted.

However, a revision is required on the manuscript (details have been provided as comments on the manuscript).

Reviewer #2: Comments to the author

The current study is about the social determinants of health influencing utilization of antibiotics leftovers in Lebanese households. The authors have found that socio-economic instability, healthcare access challenges, and experiential factors are critical in use of leftovers antibiotics among their Lebanese population which were studied.

Although the manuscript is mostly well-written in Standard English and it is clear and scientifically sound; but there are major issues according to the study objective, design and method,

1- The study does not provide any new findings or methods, and it can mostly be used as a local guidance,

2- The authors only mentioned Lebanon as their place of study which Lebanon is a country with so many cities and so much variations in their quality of life, so you had to used Cluster Sampling for choosing samples for the distribution of the questionnaire,

3- In addition having access to the internet as a inclusion criteria is a bias by its own specially in a developing country which you have not mention any statistics about internet access i.e. households with internet access. And also the platforms that you have shared the questionnaire as well can lead to limited access and biased sampling. These are the reasons that makes the study weak in representing results, since it can directly influence the knowledge of individuals about the healthcare issues in addition to effect of socio-demographic factors on access to the internet

All the above mentioned reasons question the generalizability of the study.

6. PLOS authors have the option to publish the peer review history of their article (what does this mean? ). If published, this will include your full peer review and any attached files.

**Do you want your identity to be public for this peer review?** For information about this choice, including consent withdrawal, please see our Privacy Policy .

Reviewer #1: **Yes: ** Oluwafemi J. Adewusi

Reviewer #2: No

---

## [Author Response · Author response to Decision Letter 1]

18 Aug 2025

All the concerns of the editors and reviewers were addressed and answered. The attached document entitled letter to reviewer contains all the needed information. Also, some modification requested are added to the manuscript. Thank you for your time and efforts

---

## [Decision Letter · Decision Letter 1]

14 Sep 2025

Social Determinants of Self-Medication with Leftover Antibiotics in Lebanese Households: A Cross-Sectional Study.

PONE-D-25-10954R1

Dear Dr.,

We’re pleased to inform you that your manuscript has been judged scientifically suitable for publication and will be formally accepted for publication once it meets all outstanding technical requirements.

Kind regards,

Omar El Deeb

Academic Editor

PLOS ONE

Additional Editor Comments (optional):

Reviewer #2:

Reviewers' comments:

Reviewer's Responses to Questions

**Comments to the Author**

1. If the authors have adequately addressed your comments raised in a previous round of review and you feel that this manuscript is now acceptable for publication, you may indicate that here to bypass the “Comments to the Author” section, enter your conflict of interest statement in the “Confidential to Editor” section, and submit your "Accept" recommendation.

Reviewer #2: (No Response)

2. Is the manuscript technically sound, and do the data support the conclusions?

Reviewer #2: Partly

3. Has the statistical analysis been performed appropriately and rigorously? 

Reviewer #2: No

4. Have the authors made all data underlying the findings in their manuscript fully available?

Reviewer #2: Yes

5. Is the manuscript presented in an intelligible fashion and written in standard English?

Reviewer #2: Yes

6. Review Comments to the Author

Reviewer #2: Since I have rejected the manuscript in first place, there was no need to go through the text. My comments aere so basic and there is nothing to make me change my mind

7. PLOS authors have the option to publish the peer review history of their article (what does this mean? ). If published, this will include your full peer review and any attached files.

**Do you want your identity to be public for this peer review?** For information about this choice, including consent withdrawal, please see our Privacy Policy .

Reviewer #2: No

---

## [Editor Report · Acceptance letter]

PONE-D-25-10954R1

PLOS ONE

Dear Dr. Khaddaj,

I'm pleased to inform you that your manuscript has been deemed suitable for publication in PLOS ONE. Congratulations! Your manuscript is now being handed over to our production team.

Kind regards,

on behalf of

Dr. Omar El Deeb

Academic Editor

PLOS ONE